# The TriMet_DB: A Manually Curated Database of the Metabolic Proteins of *Triticum aestivum*

**DOI:** 10.3390/nu14245377

**Published:** 2022-12-18

**Authors:** Vincenzo Cunsolo, Antonella Di Francesco, Maria Gaetana Giovanna Pittalà, Rosaria Saletti, Salvatore Foti

**Affiliations:** Laboratory of Organic Mass Spectrometry, Department of Chemical Sciences, University of Catania, Viale A. Doria 6, 95125 Catania, Italy

**Keywords:** manually curated database, mass-spectrometry-based proteomics, wheat-metabolic proteins

## Abstract

Mass-spectrometry-based wheat proteomics is challenging because the current interpretation of mass spectrometry data relies on public databases that are not exhaustive (UniProtKB/Swiss-Prot) or contain many redundant and poor or un-annotated entries (UniProtKB/TrEMBL). Here, we report the development of a manually curated database of the metabolic proteins of *Triticum aestivum* (hexaploid wheat), named TriMet_DB (Triticum aestivum Metabolic Proteins DataBase). The manually curated TriMet_DB was generated in FASTA format so that it can be read directly by programs used to interpret the mass spectrometry data. Furthermore, the complete list of entries included in the TriMet_DB is reported in a freely available resource, which includes for each protein the description, the gene code, the protein family, and the allergen name (if any). To evaluate its performance, the TriMet_DB was used to interpret the MS data acquired on the metabolic protein fraction extracted from the cultivar MEC of *Triticum aestivum*. Data are available via ProteomeXchange with identifier PXD037709.

## 1. Introduction

Rich in proteins, carbohydrates, vitamins, and minerals, wheat (*Triticum aestivum* L. ssp. *aestivum*, or “soft” wheat and *Triticum turgidum* L. ssp. *durum*, or “durum” wheat) is one of the most important foods for humans [1,2,3,4]. In particular, wheat proteins represent the main source of nutritional and sensorial properties and are the main ones responsible for the technological performance of doughs in relation to pasta making. However, for a small proportion of the population, wheat proteins can cause some adverse reactions that affect the patient’s health and quality of life [5,6,7,8]. Furthermore, knowledge of the wheat protein composition is of fundamental importance for the development of improved varieties maintaining stable yields and adapting the crop to specific regional biotic and abiotic stresses [9,10,11]. For this reason, several proteomic studies, mainly by mass-spectrometry (MS)-based methods, have been conducted to understand and characterize wheat proteins [12,13,14,15,16,17,18]. Soft wheat (*Triticum aestivum)* is structurally a hexaploid species with three homologous sets of seven chromosomes in each of the sub-genomes A, B, and D and with a large and repetitive structure [4,19,20]. The first draft sequencing of the entire genome of the *Chinese Spring* variety of common wheat has only recently been completed [21,22]. However, even today, the genomic data have not been translated into the corresponding protein sequences. Consequently, due to the lack of a comprehensive database, MS-based proteomic studies aimed at identifying wheat proteins, their content, and distribution are still facing experimental difficulties [23,24,25]. The UniProt knowledgebase (UniProtKB) represents today the most important hub for the collection of protein entries (approximately 219,750,000 protein entries), and provides the scientific community with a comprehensive, high-quality, and freely accessible resource of protein sequences and functional information [26]. The UniProtKB consists of two sections: Swiss-Prot and TrEMBL. The Swiss-Prot section contains fully manually annotated, non-redundant records (approximately 560,000 entries) [26,27]. The redundancy is minimized by merging all data from different sources into a single entry; furthermore, a protein sequence is frequently modified after comparison with ESTs (Expressed Sequence Tags), full-length transcripts, or homologous proteins from other species. On the other hand, manual checking and annotation have a disadvantage: SwissProt remains relatively small, largely incomplete, and not exhaustive. The second section, the TrEMBL (Translated EMBL), is a very large database (approximately 219,190,000 entries), which contains the automatic translation of all coding sequences (CDS) stored in the EMBL/GenBank/DDBJ nucleotide sequence database. The computer translation is not entirely perfect, so the protein entries predicted by the TrEMBL database are hypothetical and many of them are redundant and poorly or not annotated, while MS-based approaches require a genome of the species under investigation with functional annotations and a non-redundant and well-curated protein database [26]. As for the area of plant proteins, out of ~35,000 cultivated plant species, only thirty-seven have sequenced and functionally annotated genomes [28]. In fact, The Swiss-Prot section of UniProt contains approximately 40,900 reviewed entries from *Viridiplantae* (version released November 2021). However, about 50% of them belong to the two model organisms, *Arabidopsis thaliana* and *Oryza sativa*. In particular, for *Triticum aestivum,* a very limited number of reviewed protein entries are actually available due to the complexity and the large size of its genome (i.e., around 470 entries, version released November 2021). The largest part of entries of *Triticum aestivum* is reported in the TrEMBL section, which contains about 180,000 unreviewed sequences. Consequently, the high degree of redundancy, coupled with the low precision of annotations and descriptions, makes wheat proteins identification problematic. In the light of this evidence, researchers started to develop comprehensive manually curated databases to support proteomic studies [29]. For example, a manually curated database (GluPro V1.0) of gluten proteins was recently compiled, comprising 630 unique full-length protein sequences that are representative of the different types of gliadin and glutenin components [30]. In this context, this paper reports the development of a manually curated database of the metabolic proteins of *Triticum aestivum*, here called TriMet_DB (Triticum aestivum Metabolic Proteins DataBase). To test its performance, the TriMet_DB was used to interpret the mass spectrometry data acquired on the metabolic protein fraction extracted from the MEC cultivar of *Triticum aestivum*. The results were compared with those obtained from the MS data search with the Swiss-Prot.

## 2. Materials and Methods

### 2.1. The TriMet Database Compilation 

The approach carried out to compile the manually curated TriMet_DB is illustrated in Figure 1. 

In particular, to construct the database we considered, as a starting point, two lists of protein entries: (i) the complete list of *Triticum aestivum* entries reported in the Swiss-Prot section of the UniProt database (Figure 1a); (ii) the list of metabolic wheat proteins identified, using an MS-shotgun approach, in the recent work by Di Francesco et al. [31] (Figure 1b). In detail, first, the complete list of reviewed entries of *Triticum aestivum* (379 sequences from the Swiss-Prot section (release May 2021)) was downloaded from UniProt in FASTA format (Figure 1a). Then, gluten proteins were identified and eliminated manually using protein descriptions containing the terms “gluten”, “gliadin”, “glutenin”, “avenin”, and “prolamins” (Figure 1(a1)). The list of the remaining protein entries, consisting of 335 sequences, was used for the next step. In particular, each protein entry (hereafter called: “query sequence”) was BLAST (Basic Local Alignment Search Tool; https://blast.ncbi.nlm.nih.gov/Blast.cgi (accessed on 1 November 2022))-searched against the TrEMBL section of the Uniprot database (Figure 1(a2)) by using the following parameters: (a) E-threshold (statistical measure of the number of matches expected in a random database) was set to 10; (b) BLOSUM 62 (BLOcks SUbstitution Matrix) was used to evaluate the alignments [32]; (c) gaps allowed (i.e., deletions and insertions can be introduced into the alignments that are returned), and (d) max of hits reported: 1000. For each BLAST search, only *T. aestivum* sequences that shared an amino acid identity greater than or equal to 80% with the “query sequence” were downloaded, saved, and collected in a single file per protein family (Pfam) cluster prior to curation (Figure 1(a3)). Subsequently, all protein entries belonging to a Pfam cluster were compared with each other by the BLAST tool to eliminate redundant, duplicate, or fragment sequences (i.e., sequences that are part of another sequence of the cluster) (see paragraph 3.1 for details). This step made it possible to obtain, for each Pfam cluster, a list of unique and non-redundant protein entries. At this stage, the sequences were manually interrogated to remove incomplete and incorrectly annotated sequences. These entries lists were finally saved in FASTA format for the construction of the TriMet_DB (Figure 1(a4)). As reported above, to extend the collection of proteins, we also considered the list of 603 metabolic wheat proteins identified, using an MS-shotgun approach, in the recent work by Di Francesco et al. [31] (see Figure 1b). In this work, proteins were identified by searching MS data against a restricted protein database constituted by 7612 reviewed protein entries of *Triticum*, *Oryza*, *Hordeum*, *Avena*, *Secale*, *Maize*, and *Brachypodium*, downloaded from the Swiss-Prot section of the UniProt database (release July 2018). A total of 119 out of 603 protein entries came from *T. aestivum* and were, therefore, already included in the previous step performed to generate the TriMet_DB. The remaining 484 came from the other species phylogenetically related to *Triticum aestivum*. These proteins formed the second starting list to compile our database (Figure 1(b1)). In this regard, it is important to highlight that these wheat proteins have been identified thanks to the large presence in Swiss-Prot of the reviewed entries of homologous proteins from the closest related specie reported above (cross-species identification). All 484 protein entries (hereinafter referred to as: “homologue query sequences”) were BLAST-searched against the TrEMBL section of the Uniprot database using the same parameters described above (Figure 1(b2)). For each BLAST search result, only the protein entry of *T. aestivum* showing the highest identity percentage with the “*homologue query sequence*” was considered for the following steps (Figure 1(b3)). This means that each *T. aestivum* entry identified in the previous step was considered as a new “*query sequence*” and used to perform the steps described above (blasted against the UniProt database, removing duplicates, etc.; see Figure 1a). Then, all the *T. aestivum* entries identified by this approach (Figure 1(b4)) were grouped with those obtained in the first stage. Duplicate entries were eliminated, and the final list of entries was saved in FASTA format to obtain the manually curated TriMet_DB (see Figure 1(c)). The complete list of protein entries included in the TriMet_DB is also shown in Appendix A. Each entry reports the UniProt Acc. No., description, organism, gene code, status of sequence (i.e., complete or fragment), protein family, allergen name (if any), indication of last update in UniProt, and the list of the accession number of the collected entries. When the coding gene symbol was not available for wheat, the corresponding protein entry was blasted to identify the species closest related to *Triticum* present in the databases. This was performed in an attempt to address the difficult challenge of assigning a putative function, known as annotation, to these wheat proteins. Although the gold standard for annotation remains biological experimentation, and “extrapolation” of the coding gene [33] from ortholog proteins may be unreliable [34], predictions based on sequence similarity remain necessary in most of cases. 

### 2.2. Chemicals

All chemicals were of the highest purity commercially available and were used without further purification. Formic Acid (FA), dithiothreitol (DTT), ammonium bicarbonate, and iodoacetamide (IAA) were provided from Aldrich (St. Louis, MO, USA). Modified porcine trypsin was obtained from Promega (Madison, WI, USA). Water and acetonitrile (ACN) (OPTIMA^®^ LC/MS grade) for LC/MS analyses were purchased from Fisher Scientific (Milan, Italy). K_2_HPO_4_ and NaCl were obtained from Carlo Erba (Milan, Italy).

### 2.3. Sample Collection and Treatment

Grain of the bread-making wheat (*Triticum aestivum*) cultivar MEC was grown at CREA-CI, Foggia. An amount of 200 mg of wheat flour was extracted in 2 mL of extraction buffer (0.4 M NaCl, 0.067 M K_2_HPO_4_, pH 7.6) for 15 min under continuous stirring at 20 °C. The insoluble fraction was spun down at 13,523× *g*, for 15 min at 4 °C, in an Eppendorf centrifuge (Eppendorf srl, Milan, Italy). The pellet material, constituted by gliadins and glutenins, was separated and the extraction procedure was repeated twice. The supernatants containing the salt-soluble proteins (i.e., the metabolic protein fraction) were treated as [7]. Finally, 50 µg of each sample was reduced by adding 39 µg of DTT (0.01 M) in ammonium bicarbonate 0.05 M, alkylated with 92 µg of IAA (0.02 M) in ammonium bicarbonate 0.05 M, and digested by porcine trypsin (Sequencing Grade Modified Trypsin, Porcine, lyophilized, Promega) following the procedure described in [7]. To obtain a final peptide mixture concentration of 25 ng/µL, the solution was diluted with a 1% aqueous solution of formic acid. 

### 2.4. Mass Spectrometry Analysis

Mass spectrometry data were acquired following the procedure described in [7]. 

### 2.5. Database Search and Protein Identification

MS data, obtained by the triplicate LC-MS/MS runs, were processed using PEAKS X de novo sequencing software (v. 10.0, Bioinformatics Solutions Inc., Waterloo, ON, Canada). Data were searched against the manually curated TriMet_DB compiled here (constituted by 3269 entries), and against the SwissProt database restricted to the reviewed *T. aestivum* entries (379, release May 2021; hereafter called: *T. aestivum* SwissProt_DB). A database search was carried out using the parameters: (i) full tryptic peptides with a maximum of 3 missed cleavage sites; (ii) oxidation of methionine, and transformation of N-terminal glutamine and N-terminal glutamic acid residue in the pyroglutamic acid form as variable modifications; (iii) carbamidomethylation of cysteine as a fixed modification. The precursor mass tolerance threshold was 10 ppm and the max fragment mass error was set to 0.6 Da. Peptide Spectral Matches (PSMs) were validated using a Target Decoy PSM Validator node based on q-values at a False Discovery Rate (FDR) ≤ 0.1%. PEAKS score thresholds for PSMs were set to achieve, for each database search, FDR values for PSMs, peptide sequences, and proteins identified below the 0.1% value. A protein was considered identified if a minimum of two unique peptides were matched. Proteins containing the same peptides and that could not be differentiated based on MS/MS analysis alone were grouped to satisfy the principles of parsimony (or Occam’s razor) [35,36]. In a single database search, a protein was considered confidentially identified if it fulfilled both of the following requirements: (i) a minimum of two peptides with a score above the peptide filtering threshold matched; (ii) the list of the matched peptides included at least a unique peptide (i.e., a peptide with a score above the peptide filtering threshold that can be mapped to only one protein group). Then, to produce the final list of proteins identified by searching both databases, only those proteins identified at least in two out of three LC-MS/MS replicates were considered.

## 3. Results

### 3.1. TriMet_DB Development

The aim of this work was to develop an annotated, non-redundant resource of metabolic protein entries that could facilitate analysis of MS data. We collected the protein entries following the flowchart shown in Figure 1 and illustrated in the “Material and Methods” section. In particular, we used, as a point of departure, two lists of metabolic proteins as “query sequences”: (i) the 335 reviewed entries present in the Swiss-Prot database (release May 2021); (ii) the 484 proteins of the phylogenetically closest species to *T. aestivum*, identified in a recent work [31] by a shotgun approach. The entire list of “query sequence” entries is shown in Appendix A. Each “query sequence” entry was BLAST-searched against the TrEMBL section of the Uniprot database to find as many unreviewed wheat sequences as possible showing the highest percentage of sequence identity compared to the “query sequence”. This step allowed for the collection of many protein family (Pfam) clusters that were checked for redundant sequences, fragments, or duplicates. For example, the BLAST search carried out using the UniProtKB-P12299 (GLGL2_WHEAT) entry as a “query sequence” made it possible to assemble a Pfam cluster including the unreviewed UniProtDB-A0A3B5Z6A6 entry. 

The alignment, reported in Figure 2a, shows that these sequences differ only for three amino acid differences in the precursor peptide, highlighted in the red rectangle, whereas the amino acid sequence corresponding to the mature protein is the same. For this reason, this unreviewed entry was considered redundant and, therefore, not used for database compilation. Similarly, by checking another Pfam cluster, the unreviewed UniProtKB-A0A341U9I2 entry (consisting of 206 amino acids) was identified as a fragment of the UniProtKB-P17933 (RR2_WHEAT) entry (sequence length: 236 amino acids) and, therefore, discarded (Figure 2b). Finally, as an example of a duplicate sequence, Figure 2c shows the alignment between the unreviewed UniProtKB-Q332M9 entry and the UniProtKB-Q01902 “query sequence” (RT07_WHEAT). The first entry has the same amino acid sequence as the reviewed “query sequence”. Consequently, it represents a duplicate of the UniProtKB-Q01902 (RT07_WHEAT) entry and, therefore, was not included in the database. On the contrary, protein entries that differ in: i) point amino acid mutations; ii) the lack of single amino acids or of short stretches of sequences (i.e., gaps) was taken into consideration for the final compilation of the TriMet_DB.

Following this approach, 4363 sequence entries were compared using the BLAST search, 1094 proteins were discarded, whereas 3269 entries were collected to assemble the TriMet_DB FASTA file, as reported in Appendix A. 

### 3.2. Searching MS Data against the TriMet_DB and T. aestivum Swiss-Prot_DB

To evaluate the performance of the manually curated TriMet_DB, we used the MS data obtained from a shotgun analysis of an extract of the metabolic (salt-soluble) proteins of wheat flour of the MEC (see the Materials and Methods for details of the extraction procedure). Our study did not include gluten proteins (i.e., gliadins and glutenins). For gluten, refer to the Bromilow database [30]. Then, the TriMet_DB search results were compared with the results obtained by searching MS data against the *T. aestivum* Swiss-Prot_DB. The lists of proteins identified in both databases are shown in Appendix A.

In detail, 65 proteins were identified by searching in the *T. aestivum* Swiss-Prot_DB. Instead, using the manually curated TriMet_DB, 306 protein entries could be identified. As shown by the Venn diagram reported in Figure 3, 43 proteins were common to both searches, while 22 and 265 were identified exclusively by *T. aestivum* Swiss-Prot_DB and TriMet_DB, respectively. Although the exclusive identification of many entries in the TriMet_DB was expected, the 22 proteins detected only in the *T. aestivum* Swiss-Prot_DB might seem surprising because these protein entries are also included in the manually curated database assembled here. To explain this apparent inconsistency, there are some important aspects of the current findings that are worth exploring in the discussion paragraph.

## 4. Discussion

To understand why 22 proteins, appear to be identified exclusively in Swiss-Prot_DB, although they were also included in the TriMet_DB, it is important to consider that database searching of the MS/MS spectra provides matches to peptide sequences [37], not proteins. Proteins are indirectly identified using the corresponding peptides. The most important requirement is that the analyzed peptide sequences must be contained in the protein sequence database investigated. As a result, a protein is identified if it is present in the database. For simple organisms where most peptides map uniquely to a protein, this is straightforward. However, in higher eukaryotes using these matches to identify which proteins were present in the original sample may be very difficult because many of these peptide sequences can be assigned to more than one protein. Consequently, as peptide sequences must be assembled into a list of identified proteins, when a peptide can be mapped to multiple proteins, this leads to the problem of protein inference [38]. Although the protein inference problem is not the focus of this paper, it is important to underline that two strategies may be applied to compile the final report of the identified proteins [39]. In fact, a report created from database search results can take a maximal or minimal approach to listing the identified proteins. The maximal list, also named anti-Occam’s razor, is based on the maximal explanatory set of proteins, where any protein that is matched by at least one identified peptide will be included in the reported protein list. The maximal approach, which is strongly discouraged by the journal guidelines, may only be useful for smaller searches, where manual inspection will be used to decide which proteins were truly present in the original sample. The minimal list, which is the common strategy used by the most post-processing MS data (i.e., the PEAKS X software used here, v. 10.0, Bioinformatics Solutions Inc., Waterloo, ON, Canada) is the smallest set of database entries that accounts for all identified peptides. The mechanism for selecting a minimal list is often described as Occam’s razor or the principle of maximum parsimony. Using this strategy, proteins identified from the same set of peptides and which cannot be differentiated based on MS/MS analysis alone are usually pooled. On the other hand, proteins that are not supported by at least a unique peptide are discarded. Figure 4 shows the fundamental ways of the Occam’s razor strategy, in which proteins can be related through shared correspondences. 

MS search algorithms, using this strategy, group proteins that span the same set or a subset of peptides, and consider as identified only the protein (classified as “top protein”) supported by the unique peptides (i.e., belonging only to the top protein of the group). Conversely, protein entries of the group that are identified by a subset of the common peptides supporting the “top protein” will be discarded (see Figure 4; case a). On the other hand, some proteins could be identified through a group of peptides shared with other proteins entries, but not by unique peptides (see Figure 4; case b). These proteins, also called “intersection proteins”, are, therefore, according to the principle of parsimony, excluded from the report. In light of these brief considerations, as reported above, we considered a protein as confidentially identified if it met both of the following requirements: (i) a minimum of two peptides with a score above the filtering threshold of the matched peptide; (ii) the list of the matched peptides included at least one unique peptide (i.e., a peptide with a score above the peptide filtering threshold that can be mapped to a single protein group). Actually, 15 of the 22 proteins apparently identified only in *T. aestivum* Swiss-Prot_DB were also detected by the TriMet_DB search, but discarded because they were identified from a list of peptides that was a subset of peptides supporting the “top proteins” of the groups (Figure 4, case a). The complete list of these 15 proteins and their related identified peptides are reported in Appendix A. It is important to underline that all these “top proteins” are unreviewed entries (i.e., included only in the TrEMBL database) and, therefore, not identifiable by searching the *T. aestivum* Swiss-Prot_DB. In this database, for example, the ATP synthase subunit alpha (Acc. No. P12862, ATPAM_WHEAT) was identified by eight peptides, all uniquely related to this entry. On the other hand, the same group of peptides allowed the detection of this protein entry also by searching the manually curated database. However, in the latter search, this group of peptides also allowed the identification of the unreviewed UniProtKB-A0A3B6RD12 (A0A3B6RD12_WHEAT) entry. The identification of the A0A3B6RD12_WHEAT entry was also supported by the matching of two additional peptides (see Figure 5). 

Interestingly, these two peptides show a point difference with respect to the P12862 entry and, therefore, are attributable only to the unreviewed sequence. In light of these considerations, the identification of the unreviewed entry explained all the peptides observed and, by the principle of maximum parsimony, the P12862 entry can be removed from the report. It is important to point out that both these proteins are ATP synthase subunit alpha, differing only in a few point substitutions (see Figure 5). The first entry (Acc. No. P12862) is a reviewed sequence and, therefore, present in the Swiss-Prot database; the second (Acc. No. A0A3B6RD12) is an unreviewed entry and, consequently, reported only in the TrEMBL section. Finally, this example highlights that searching a curated database allows for more accurate protein identification, including any protein isoforms not yet represented in the Swiss-Prot database. Furthermore, among the 22 protein entries recognized exclusively in the *T. aestivum* Swiss-Prot_DB, seven of them (i.e., the entries with Acc. No. F8RP11, P12782, O64393, P93594, P02277, P32112, and P62786) were not listed in the final report of the TriMet_DB search, because they did not satisfy one of the two requirements aforementioned. Particularly, the F8RP11 entry (Hsp70-Hsp90 organizing protein) and P12782 entry (Phosphoglycerate kinase, chloroplastic) were identified with a single peptide with a significance score above the set threshold, and, therefore, did not satisfy the first requirement. The other five proteins (O64393, P93594, P02277, P32112, and P62786) instead represent a classical example of “intersection protein” (see Figure 4, case b), not supported by unique peptides. As an example, Figure 6 shows the Occam’s razor strategy applied to assemble fourteen analyzed peptide sequences into a list of identified proteins by searching both the Swiss-Prot (identified entries sp|O64393 and sp|O64392) and the manually curated TriMet database (identified entries sp|O64392 and the unreviewed tr|A0A3B6G229). 

In detail, thirteen out of the fourteen peptide sequences analyzed allowed the identification of the sp|O64393, supported by the unique peptide N. 6 (amino acid sequence: YGWTAFCGPAGAHGQAACGK), and of the sp|O64392, supported by the unique peptides N. 4 and N. 7. Instead, the analyzed peptide sequence N.8 (amino acid sequence: VTNPATGAQVTAR) remains unmatched because the unreviewed tr|A0A3B6G229 entry is not included in Swiss-Prot. In contrast, by searching the curated TriMet database, the fourteen peptide sequences analyzed allow the assembly of a list of identified proteins, including the sp|O64392 and tr|A0A3B6G229 entries, which are supported by unique peptides. Peptides N. 4 and 7 support the sp|O64392, whereas peptide N. 8 is unique of the tr|A0A3B6G229 entry. Instead, the sp|O64393 entry represents a subset of the combined matches to the previous two entries (i.e., it is an “intersection protein”), and is, therefore, deleted from the report. It is important to note that these three proteins share more than 95% of the primary structure (Figure 7), differing by a few point substitutions. 

Therefore, this example highlights once again that searching in the curated database can allow the obtaining of not only a more accurate protein identification, but more generally a deeper description of the protein fraction under investigation, because many analyzed peptide sequences that remain uninterpreted by searching Swiss-Prot_DB are instead assigned because they are contained in the curated database. 

## 5. Conclusions

Proteomic analysis has among its goals the identification of as many proteins as possible with high confidence. The list of the identified proteins can then be classified into the putative molecular function and biological process categories by Gene Ontology analysis, and these results can be used to address the biological issues relevant in the study under consideration. Therefore, the choice of the protein sequence databases plays a fundamental role in mass-spectrometry-based proteomics workflows currently applied and has a strong impact on the results of the search. As an example, if not all proteins in the sample are present in the database, peptides from such unexpected proteins, and, therefore, the related MS spectra, can be matched incorrectly to other proteins in the database, which results in false positive identifications. The proteomics of wheat is currently facing experimental difficulties, because the information about wheat genome sequence has not been converted in the protein sequence and a complete database of wheat proteins is still lacking today. The UniProt_DB is the most important hub for the collection of wheat protein entries, but it contains just under 400 annotated, non-redundant, and reviewed entries (i.e., the Swiss-Prot section) from the *T. aestivum* specie. Most of the *T. aestivum* entries are included in the TrEMBL section, which contains approximately 143,000 “unreviewed”, poorly, or un-annotated sequence entries. Consequently, MS-based proteomic studies aimed at identifying wheat proteins are currently challenging, and tailoring the database is advisable. As an example, protein entries of closely related species can also be included to implement the database, because traits of the sequence are conserved between multiple species [7,31]. Conversely, the TrEMBL database search leads to many redundant results, which require long and tedious manual checking of the identified proteins. Moreover, the list of identified proteins in this circumstance mainly includes non-annotated entries (i.e., “hypothetical” and “uncharacterized” proteins). This impedes, in principle, our understanding of the findings at the molecular level, the most important goal of biological research. Here, we developed the TriMet_DB, containing 3269 entries of metabolic proteins of wheat. The approach carried out to develop it allowed the organizing (i.e., the classification of the protein family and the attribution of the gene code) of all the selected protein entries, including the high number of hypothetical and uncharacterized entries selected from the section TrEMBL. To evaluate its performance, the TriMet_DB was used to interpret the mass spectrometry data acquired on the metabolic protein fraction extracted from the MEC cultivar of *T. aestivum*, and results were compared with those obtained by searching MS data against the Swiss-Prot database (and using the taxonomy restriction to *T. aestivum*). Overall, the results obtained by the investigation of the protein content of the MEC wheat bread-making cultivar show that the use of a curated database allows the achievement of a more in-depth description, including (i) possible protein isoforms not yet represented in the Swiss-Prot database, and (ii) an easier Gene Ontology analysis. The manually curated TriMet_DB was generated in FASTA format so that it can be read directly by programs used to interpret the mass spectrometry data. It can be freely download by the Appendix A section of the present publication. Furthermore, the complete list of entries included in the TriMet_DB is also a freely available resource reported in Appendix A, and includes for each protein the description, the gene code, the protein family, and the allergen name (if any). Future work will focus on implementing an automated algorithm to expand the database, which, up until now, is certainly not complete. It is worth noting that the TrEMBL database contains more sequence entries (partly also annotated) that probably do not group into any of the clusters described here, and, therefore, will represent additional proteins that are not yet part of the clustered TriMetDB database. Moreover, the integration of these sequences can facilitate the identification and the functional study of metabolic proteins of wheat.

## Figures and Tables

**Figure 1 nutrients-14-05377-f001:**
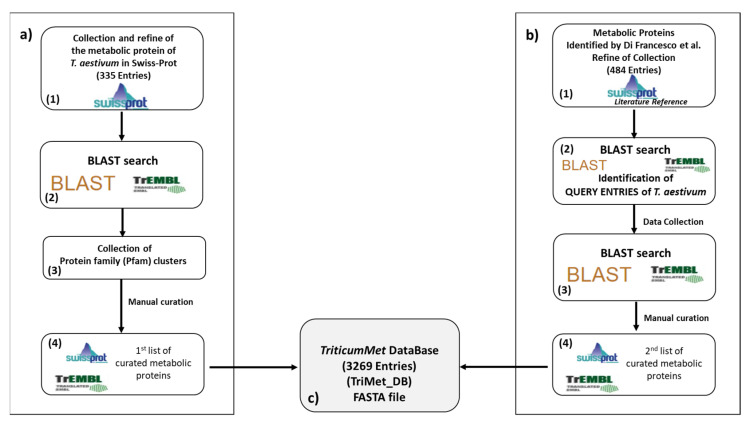
Scheme of the approach carried out to compile the manually curated TriMet Database. (**a**) Scheme of the approach that used as a starting point the complete list of *Triticum aestivum* entries reported in the Swiss-Prot section of the UniProt database; (**b**) scheme of the approach that used as a starting point the list of metabolic wheat proteins identified, by means of a MS-shotgun approach, in the recent work of Di Francesco et al. (Di Francesco et al., 2019) [31] (**c**) Number of sequence entries included in the TriMet_DB FASTA file.

**Figure 2 nutrients-14-05377-f002:**
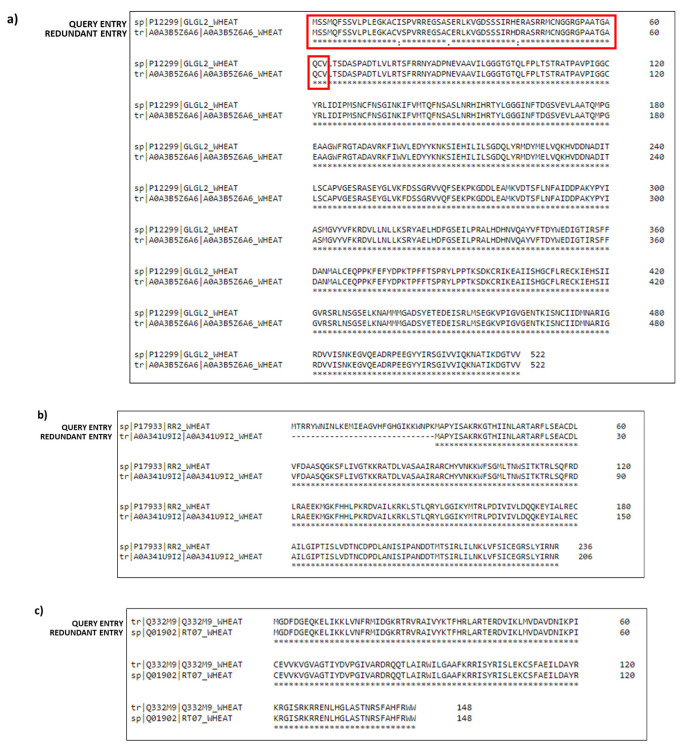
Examples of sequences discarded because of redundancy. For each example, the alignment of the query and the redundant entries is reported. Alignment displays the consensus symbols, which denote the degree of conservation observed in each column. The symbol “*” means that the amino acids in that column are identical in both sequences. The symbol “:” means that conserved substitutions are observed. The symbol “.” means that semi-conserved substitutions are observed. (**a**) Example of two sequences differing only for a point amino acid mutation in the peptide signal (reported in the red rectangle); (**b**) example of fragment sequence; (**c**) example of duplicate sequences.

**Figure 3 nutrients-14-05377-f003:**
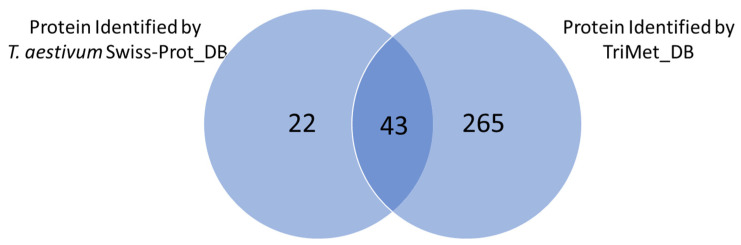
Venn diagram reporting the proteins identified using two different databases, *T. aestivum* Swiss-Prot_DB and TriMet_DB.

**Figure 4 nutrients-14-05377-f004:**
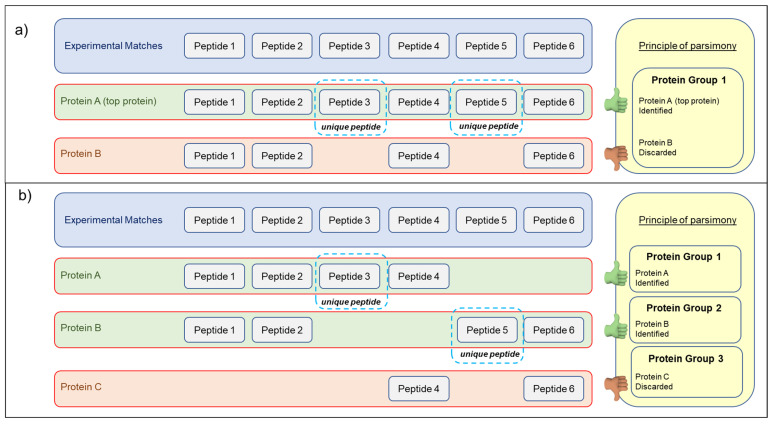
The fundamentals of Occam’s razor strategy (principle of maximum parsimony) in Proteomics, by which proteins can be related through shared matches, and used to create the final report of the identified proteins. (**a**) B is a subset of protein A. Protein B may be present in the sample, but there is no evidence for this, so parsimony may be dropped from the report. (**b**) Protein C can be considered as an “intersection protein” (i.e., a subset of the combined matches to A and B). By Occam’s razor, it may be discarded from the report because it is not identified by unique peptides.

**Figure 5 nutrients-14-05377-f005:**
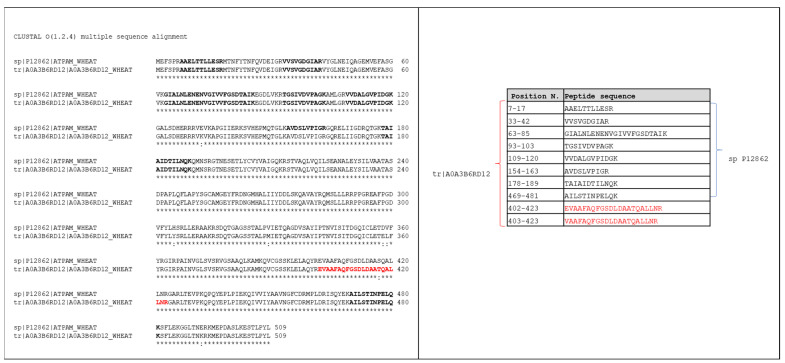
Alignment of the tr|A0A3B6RD12 and sp|P12862 entries. The characterized sequence traits are reported in bold (see also the table on the right). In bold red is the reported sequence trait 402–423 characterized by the identification of the two unique peptides (reported in red in the table on the right) related only to unreviewed entry tr|A0A3B6RD12.

**Figure 6 nutrients-14-05377-f006:**
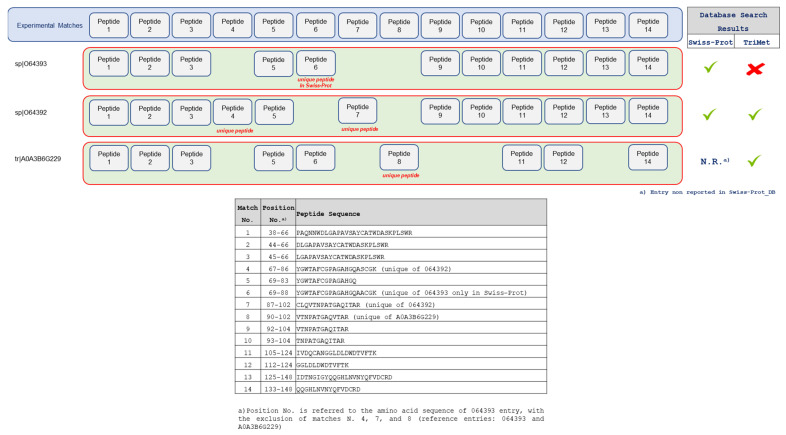
Occam’s razor strategy applied to assemble the fourteen analyzed peptide sequences (showed in the table on the bottom of Figure 6) into a list of identified proteins by searching both the Swiss-Prot and the manually curated TriMet database. See the main text for details.

**Figure 7 nutrients-14-05377-f007:**
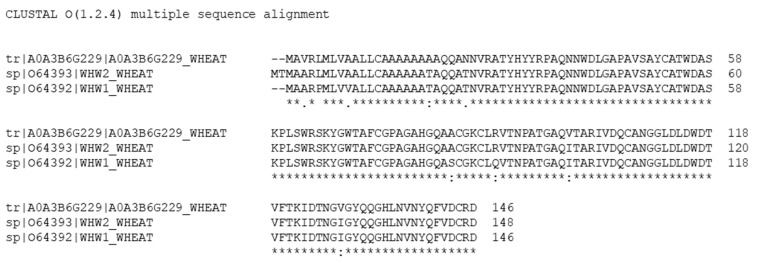
Alignment of the tr|A0A3B6G229, sp|O64393, and sp|O64392 entries. Alignment displays the consensus symbols, denoting the degree of conservation observed in each column.

## Data Availability

Data are available via ProteomeXchange (www.proteomexchange.org, accessed on 1 November 2022) with identifier PXD037709.

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
