# Peer review of "The TriMet_DB: A Manually Curated Database of the Metabolic Proteins of Triticum aestivum"

_nutrients, 2022, doi:10.3390/nu14245377_

Round 1
Reviewer 1 Report
Comments and Suggestions for Author:
This manuscript “TriMetDB: a manually curated database of the metabolic proteins of Triticum aestivum” describes an elaborate approach to cluster and annotated a large collection of wheat protein sequence accessions of the UniProt / TrEMBL database on the basis of sequence homologies with peer reviewed accessions in UniPROT / SwissProt database.
This approach and the resulting data (FASTA data and annotation table) are relevant and helpful for interpreting proteomics data with a focus on the metabolic proteins of wheat. The report is excluding the gluten (i.e. gliadin and glutenin) proteins, as these have been reported elsewhere (ref [28]).
My major considerations for this manuscript are:
· Normally I would consider this type of manuscript more relevant to a journal in the proteomics field ( Proteomics, Journal of Proteomics , J of Proteome Research), not so much for Nutrients. As the approach and results are typically relevant for proteomics interpretation. However, in case of this special issue "'Protein Matters and Proteins Matter': Proteomics and Peptidomics in Nutrition and Health" it may be of some relevance. I leave the decision on this for the Guest editor (obviously).
· For the reader of “Nutrients” it may not be directly clear that the used extraction process, and also the data analysis only focusses on the salt soluble proteins. Some lines in discussion section should mention that gluten (and other non-salt soluble proteins) are not included in this study (not in the database and not in the MS analysis). For gluten the referral to the Bromilow database [28] is relevant.
· The manuscript describes the clustering of proteins from the non-reviewed section of UniProt ( TrEMBL) on the basis of the reviewed entries (from wheat or its close relative cereals). This transfers the functional annotation to the non-reviewed proteins. This is relevant for interpretation/ annotation of results. However, the manuscript describes the use of the FASTA database as a search space for proteomics data analysis. I do not fully agree in this reasoning. As the authors correctly state in their discussion, peptides will be only identified on basis of presence of their sequence in the database. However, the described database is not complete. As it is based on the 335 triticum and 484 (other related species) SwProt sequences. The TrEMBL database contains more proteins (partly also annotated) that will not group into any of the described clusters. My approach would be to search the complete Triticum subset of sequences in UNiProt (SwPr and TrEMBL) and subsequently try to link the identified proteins into the clusters as they are described here. In this way, additional proteins may be identified that are not yet part of the clustered TriMetDB database. A discussion on this approach would be appreciated.
· I think that the arguments in line432-433 are not so relevant anymore. reference [38] is from 2007…
· The description of the “protein inference” problem is very valuable. This problem is really an issue, especially in cereal proteomics (with the multitude of isoforms and homologous protein sequences). Having a good description of this issue, like here, provides more attention and insight of this matter for interpretation of proteomics data. Although the discussion is lengthy, it is worth it. But maybe not so relevant for “Nutrients” audience.
· I suggest to transfer table 1, being a table of 21 pages to the supplemental section. It is too long for the main text. And it is of more use to the readers if it is in a table (excel) format available.
· The TriMetDatabase is provided as a .doc (Word) format. It should be provided as a plain .txt or .fasta format, which can be read into MS search software.
· The database contains 644 accessions Table S1, column A) and 2982 collected entries (column I). This does not add up to (335 +484)= 819 accessions (query sequences) and 3269 collected entries in the database. I think these will be unique entries? But this is not described. Please provide the same info on the remaining (collected) entries into table S1.
Minor corrections:’
- Line 32:knowledge of the wheat protein composition is of ..
- Line 36: references 12-15 only refer to work of the authors themselves. There is a multitude of publications on wheat proteomics also from others:
o I suggest to add a.o.references to :
o Vincent, D.; Bui, A.; Ram, D.; Ezernieks, V.; Bedon, F.; Panozzo, J.; Maharjan, P.; Rochfort, S.; Daetwyler, H.; Hayden, M. Mining the Wheat Grain Proteome. Int. J. Mol. Sci. 2022, 23, 713. https://doi.org/ 10.3390/ijms23020713
o Afzal, Muhammad, …., Longin, C. F. H. High-resolution proteomics reveals differences in the proteome of spelt and bread wheat flour representing targets for research on wheat sensitivities. https://doi.org/10.1038/s41598-020-71712-5
o Azadeh Fallahbaghery , Wei Zou , Keren Byrne , Crispin A Howitt , Michelle L Colgrave. Comparison of Gluten Extraction Protocols Assessed by LC-MS/MS Analysis. J Agric Food Chem. 2017 Apr 5;65(13):2857-2866. doi: 10.1021/acs.jafc.7b00063
- Line98: (Di Francesco et. Al 2019) should be removed , as [29] is already the reference.
- Line 122 and 208: same as above
- Line 189: .. was carried out the using the …
- Line 230: .. signal peptide..
- Line 246: “internal traits of sequences” should be : “short stretches of sequences” (gaps)
- Line 249-250: remove redundant text : “and a total of….. Suppl Table S1” this is repeated.
- Line 260: move table 1 to Suppl info..
- Figure 5 right panel: the accolade brackets are too long .. adjust it
- Line 383 fourth-teen should be fourteen
- Line 406: .. one can be obtain ..
- Line 455: presents
- Line 508: doi:10.3389/fpls.2016.02020
- Line 589: R., E.S. … looks like something is missing here
- Line 591: journal and doi are missing
Author Response
Dear Reviewers,
we wish to thank you for your helpful comments and suggestions.
Point-by-point responses to the comments are given below.
Reviewer N.1
- For the reader of “Nutrients” it may not be directly clear that the used extraction process, and also the data analysis only focusses on the salt soluble proteins. Some lines in discussion section should mention that gluten (and other non-salt soluble proteins) are not included in this study (not in the database and not in the MS analysis). For gluten the referral to the Bromilow database [28] is relevant.
- Some lines were added in the first part of the paragraph 3.2 to highlight that our study was focused on the salt-soluble proteins (i.e the metabolic protein fraction) and does not include gluten proteins
- The manuscript describes the clustering of proteins from the non-reviewed section of UniProt (TrEMBL) on the basis of the reviewed entries (from wheat or its close relative cereals). This transfers the functional annotation to the non-reviewed proteins. This is relevant for interpretation/ annotation of results. However, the manuscript describes the use of the FASTA database as a search space for proteomics data analysis. I do not fully agree in this reasoning. As the authors correctly state in their discussion, peptides will be only identified on basis of presence of their sequence in the database. However, the described database is not complete. As it is based on the 335 triticum and 484 (other related species) SwProt sequences. The TrEMBL database contains more proteins (partly also annotated) that will not group into any of the described clusters. My approach would be to search the complete Triticum subset of sequences in UNiProt (SwPr and TrEMBL) and subsequently try to link the identified proteins into the clusters as they are described here. In this way, additional proteins may be identified that are not yet part of the clustered TriMetDB database. A discussion on this approach would be appreciated.
- We thank the reviewer for this comment, which is absolutely correct. In fact, as a part of our work for developing the TriMetDB database, we have already tried the approach suggested. The result was a list of more than a thousand proteins, which is almost impossible to check, as this procedure is time-consuming when done manually. In fact, to obtain the final list of 3269 collected entries, included in the TriMet_DB we manually controlled about 5000 blast searches. In collaboration with computer science colleagues, we are currently developing a software for automated database search. This software will perform the suggested approach and will also be required for a periodic update of the TriMetDB database. To clarify this aspect a sentence was added in the paragraph “Conclusion” of the main text.
- I think that the arguments in line 432-433 are not so relevant anymore reference [38] is from 2007.
- The sentence and the corresponding reference [38] were deleted
- The description of the “protein inference” problem is very valuable. This problem is really an issue, especially in cereal proteomics (with the multitude of isoforms and homologous protein sequences). Having a good description of this issue, like here, provides more attention and insight of this matter for interpretation of proteomics data. Although the discussion is lengthy, it is worth it. But maybe not so relevant for “Nutrients” audience.
- We perfectly agree with the reviewer. The “protein inference” problem is an issue, especially in cereal proteomics. Although the discussion of this issue is lengthy, we consider that it may be, in general, relevant. We think that a shortened discussion might be not exhaustive to explain this problem. Therefore, we prefer don’t reduce this paragraph.
- I suggest to transfer table 1, being a table of 21 pages to the supplemental section. It is too long for the main text. And it is of more use to the readers if it is in a table (excel) format available.
- We agree with the reviewer. The Table 1 was removed from the main text, included in the Supplementary Material, and renamed Table S2.
- The TriMetDatabase is provided as a .doc (Word) format. It should be provided as a plain .txt or .fasta format, which can be read into MS search software.
- The TriMetDatabase is now provided as a .text format.
- The database contains 644 accessions Table S1, column A) and 2982 collected entries (column I). This does not add up to (335 +484) =819 accessions (query sequences) and 3269 collected entries in the database. I think these will be unique entries? But this is not described. Please provide the same info on the remaining (collected) entries into table S1.
- We would like to thank the reviewer because the description of the Table was effectively unclear and incomplete. In the development of the TriMet_DB we used two lists of metabolic proteins, as starting point: i) the 335 reviewed entries present in the Swiss-Prot database and ii) the 484 proteins of the phylogenetically closest species to T. aestivum. A preliminary comparison, by BLAST search analysis, of these two protein groups allowed the identification of 175 entries (belonging to the second group) that were discarded because already saved and collected in a file per protein family (Pfam) cluster. Therefore, the column A contains the remaining entries after this check (i.e. 644 proteins). The column “Collected entries” reports a total of no-redundant 2625 entries. We now added this information in the Table header.
Minor corrections:
- Line 32: knowledge of the wheat protein composition is of ..
- Modified as suggested by the reviewer.
- Line 36: references 12-15 only refer to work of the authors themselves. There is a multitude of publications on wheat proteomics also from others:
- The bibliographic references were added as suggested by the reviewer.
- Lines 98, 122, and 208: (Di Francesco et. Al 2019) should be removed , as [29] is already the reference.
- The sentence “Di Francesco et. al, 2019” was removed.
- Line 183: .. was carried out using the …
- The sentence was modified as suggested by the reviewer.
- Line 230: .. signal peptide..
- The sentence was modified in “precursor peptide”.
- Line 246: “internal traits of sequences” should be : “short stretches of sequences” (gaps)
- The sentence was modified as suggested by the reviewer.
- Line 249-250: remove redundant text : “and a total of….. Suppl Table S1” this is repeated.
- The sentence was modified as suggested by the reviewer.
- Line 260: move table 1 to Suppl info..
- The Table 1 was included in the supplementary Section and renamed Table S2.
- Figure 5 right panel: the accolade brackets are too long .. adjust it
- Figure 5 was modified.
- Line 383 fourth-teen should be fourteen
- The text was modified as suggested by the reviewer.
- Line 406: .. one can be obtain ..
- The text was modified.
- Line 455: presents
- The text was modified.
- Line 508: doi:10.3389/fpls.2016.02020
- DOI was added.
- Line 589: R., E.S. … looks like something is missing here
- The text was modified.
- Line 591: journal and doi are missing
- The text was modified.

Reviewer 2 Report
Line 35, sentence ending [12-15]: Add a link ‘in terms of generating a protein database’ and explain the challenge posed by the genomic data.
Line 98: Previous the recent work of Di Francesco et al. 92 (Di Francesco et al., 2019) [29] – this has associated data set Project PXD014449? This should be cross-referenced?
Lines 170, 171 Section 2.3 – it’s usual to list the concentration of DTT and IAA in millimolar – please do this.
Lines 173-174: ‘To obtain a final concentration of 25 ng/µL, the solution 173 was diluted with 5% aqueous solution of formic acid.’ Does this relate to the trypsin or peptide concentration? – this is unclear. 5% is high concentration of formic acid – it can lead to formylation + 27.994915 Da – was this tested/accounted for? Similarly, the modification of carbamidomethylation should be assessed not only for cysteines but potentially other residues due to side reactions.
Line 182: Database search was conducted using the parameters described in [29]. These should be listed in the Method for ease of assessment.
The proteomic dataset could not be assessed PXD037709, require access.
Author Response
Dear Reviewers,
we wish to thank you for your helpful comments and suggestions.
Point-by-point responses to the comments are given below.
Reviewer N.2
- Line 35, sentence ending [12-15]: Add a link ‘in terms of generating a protein database’ and explain the challenge posed by the genomic data.
- We are not sure we understood the referee's suggestion. We have included further bibliographic references, also in accordance with the comments of referee 1.
- Line 98: Previous the recent work of Di Francesco et al. 92 (Di Francesco et al., 2019) [29] – this has associated data set Project PXD014449? This should be cross-referenced?
- Yes, this is the data set associated with that work. The link of the data set project PXD014449 was added in the text.
- Lines 170, 171 Section 2.3 – it’s usual to list the concentration of DTT and IAA in millimolar – please do this.
- The text was modified as suggested by the reviewer
- Lines 173-174: ‘To obtain a final concentration of 25 ng/µL, the solution 173 was diluted with 5% aqueous solution of formic acid.’ Does this relate to the trypsin or peptide concentration? – this is unclear. 5% is high concentration of formic acid – it can lead to formylation + 27.994915 Da – was this tested/accounted for? Similarly, the modification of carbamidomethylation should be assessed not only for cysteines but potentially other residues due to side reactions.
- Thank you for your comment. Usually we use a solution with 1% of formic acid, we made a typing error. The text was modified. Moreover, we also checked for the possible carbamidomethylation in other residues than cysteines, but no results were obtained.
- Line 182: Database search was conducted using the parameters described in [29]. These should be listed in the Method for ease of assessment.
- Database parameters are now included in the text
- The proteomic dataset could not be assessed PXD037709, require access.
- The reviewer account details were included in the “Cover Letter” file in the first manuscript submission. Particularly, reviewer account details are the following:
Username: [email protected] ; Password: t2J0CeZz.
